# Biocrude Production via Non-Catalytic Supercritical Hydrothermal Liquefaction of *Fucus vesiculosus* Seaweed Processing Residues

**Lukas Jasiūnas** [1],*  , **Thomas Helmer Pedersen** [2]  **and Lasse Aistrup Rosendahl** [2]

1   Department of Organic Chemistry, Kaunas University of Technology, LT-50254 Kaunas, Lithuania
2   Department of Energy Technology, Aalborg University, 9220 Aalborg Øst, Denmark;
    thp@et.aau.dk (T.H.P.); lar@et.aau.dk (L.A.R.)
*   Correspondence: lukas.jasiunas@ktu.lt

**Abstract:** The potential of using cold water brown macroalgae *Fucus vesiculosus* for biocrude production via non-catalytic supercritical hydrothermal liquefaction (HTL) was studied. Demineralization, residue neutralization, and high value-added product (alginate and fucoidan) extraction processes were carried out before using the biomass for HTL biocrude production. Acid leaching was carried out using three demineralization agents: distilled water, dilute citric acid solution, and the diluted acidic aqueous by-product from a continuous HTL pilot facility. Alginate was extracted via $H_2SO_4$ and $NaCO_3$ bathing, and fucoidan was extracted using $CaCl_2$. Experimental data show that none of the leaching agents was greatly efficient in removing inorganics, with citric acid leaching with extensive neutralization reaching the highest ash removal efficiency of 47%. The produced 6 sets of biocrudes were characterized by elemental and thermogravimetric analyses. Short (10-min retention) HTL and the extent of leaching residue neutralization were also investigated. Highest biocrude yields were recorded when liquefying non-neutralized citric acid leaching, alginate, and fucoidan extraction residues. On the other hand, thermochemical conversions of short retention time HTL, full neutralization extent, and baseline (dried raw macroalgae) biomass performed worse. Specifically, the highest biocrude yield of 28.2 ± 2.5 wt.% on dry ash-free feedstock basis was recorded when liquefying alginate extraction residues. Moreover, the highest energy recovery of 52.8% was recorded when converting fucoidan extraction residues.

**Keywords:** residue valorization; hydrothermal liquefaction; biorefinery; macroalgae; value-added products

## 1. Introduction

The transportation sector is engaging with innovation to address societal concerns over climate change. Numerous upcoming technologies are posed to significantly diminish our dependence on fossil fuels. Among the rapidly developing technologies are hydrogen-, electricity-, and electrofuel-based alternatives. Given the imminent transitional period, intermediary fuels will undoubtedly play a critical role to gradually transform the current well-established infrastructure of liquid fuels. This is where advanced biofuels come into play and supply for this demand. Biomass, given adequate management, sustainable cultivation, and timely integration, should be the key precursor for several types of fuels—a significant fraction of future energy portfolio. Due to the wide abundance and short life cycles, biomass promises potential for a more sustainable world, one where we are able to lower anthropogenic $CO_2$ emissions drastically.

Macroalgae, also known as seaweed, constitute numerous large multicellular algae species. These seabed dwelling plants grow in coastal marine areas, and can be harvested at depths less than 50 m below sea level. The environments in such ecosystems are conveniently next to invariant in terms of temperatures and salinity, facilitating continuous

growth all year round, albeit not constant in growth rate [1]. Seasonal solar irradiance variance plays a major role in dictating growth rates and the chemical composition of the resultant macroalgal biomass. Previous studies have shown that brown seaweed (e.g., *Saccharina*, *Undaria*, *Ecklonia*, or *Sargassum*) are characterized by growth rates of 3.3–11.3 kg dry weight/m$^2$ per year [1]. This corresponds to harvest potentials between 2–10 dry tons/ha per year in Danish waters [2]. In fact, brown seaweeds can have a maximum energy yield of more than 45% throughout a single growing period. Such a value is significantly greater when compared to yields of most types of terrestrial biomass (e.g., lignocelluloses: 20–25%, energy crops: 30–35%). Such high productivity rates show a high potential for growing this biomass commercially [3].

Despite stagnant conditions locally, macroalgae are known to vary greatly in terms of chemical composition. Energy storage carbohydrate (e.g., laminaran and mannitol) fractions depend heavily on harvest seasonality, as the plants accumulate and release the compounds throughout the lighter and darker seasons, respectively [1]. Ash content can also vary greatly [1,4]. For instance, brown seaweeds harvested early in spring typically contain high amounts of alginate, proteins, and ash but low concentrations of the other types of carbohydrates [5]. However, upon receiving more light, the photosynthetic activity of the algae surges—the plants produce higher amounts of sugars, whereas the relative amounts of alginate, proteins, and ash drop [5].

Alginate, a linear polysaccharide abundant in free hydroxyl and carboxyl groups, and fucoidan, a fucose-containing sulfated polysaccharide, are amongst many algal compounds that have generated great interest in the scientific community over recent years [6,7]. Specifically, these two types of natural polysaccharides are valued for their applicability in medicinal and pharmaceutical fields. Although the properties of these value-added compounds vary depending on the chosen extraction methods and subsequent modification procedures, the processes typically result in a residual biomass by-product.

Using macroalgae as feedstock for energy production is not new, with many research groups worldwide studying the potential to make use of this marine resource. The tested technologies are of biochemical or thermochemical nature, where the biomass is converted to energy carriers. The high moisture content and high amounts of low melting point alkali and alkaline earth metals present in the biomass renders it a poor choice for direct combustion. Typically, a particular pathway is chosen based on the desired state of the output fuel. Multiphase, except for solid, fuel precursor production has been demonstrated using seaweed.

Hydrothermal liquefaction (HTL), unlike anaerobic digestion, is capable of fast production of high-quality fuel precursors, typically being carried out in a matter of minutes to tens of minutes [8]. Alkaline homogeneous catalysts are often employed, thus despite the overall high levels of ash, the metals of alkali nature present in the biomass are hypothesized to potentially improve the conversion. Reaction media of elevated pH levels are typically used, as such conditions lead to decreased formation of residual solids, whereas the gases are pushed towards repolymerization. Anastasakis et al. found that in the HTL of macroalgae, experiments in which no external catalyst was added yielded the highest amounts of biocrude [9]. In terms of quality, HTL biocrude is capable of reaching high energy densities, often equivalent to at least 70% of that of fossil crude [9]. The presence of heteroatoms in the biomass is one of the reasons why it is difficult to achieve high energy content in nontreated HTL biocrude. In macroalgae, nitrogen and sulfur are derived from proteins and sulphated carbohydrates, respectively, while all major groups of polysaccharides contain copious amounts of oxygen. When compared to lignocellulose products, algae-derived HTL biocrudes typically are more contaminated due to the high nitrogen and sulfur contents in the initial feedstock [10].

HTL experiments have been carried out with macroalgal feedstock recently [10–13]. However, alginate and fucoidan—two high-value compound extraction residues—have not yet been exposed to such thermochemical conversion conditions. On the other hand, energetic utilization via anaerobic digestion [14], pyrolysis [15], and hydrothermal car-

bonization [16] has been studied with such algal residues. Since HTL can successfully process sewage sludge and compost, two low-value wet material streams, which suggests that seaweed residues could be susceptible as well [17]. Researchers have recently studied two-stage HTL and co-liquefaction as a means to boost yields of macroalgae-derived biocrude [18,19]. This study focuses on utilizing residual material streams generated at macroalgal factories (i.e., alginate and fucoidan extraction residues). Taking circular economy goals into consideration, such modern bio-refineries could expand and produce both high-value products and HTL biocrudes at high, continuous production capacities. This study was carried out to test whether alginate and fucoidan residues can be effectively liquefied into high-quality biocrudes intended for downstream upgrading and refining of the eventually drop-in quality fuel. Additionally, the study included investigating the effects raw macroalgae demineralization, considering that the post-processed macroalgal biomass is hypothesized to contain copious amounts of inorganics. Finally, extensive neutralization of leaching residues was carried out to test for its necessity, and a shorter reaction time was investigated for potential benefits in supercritical HTL of low-value residual seaweed biomass.

## 2. Results

### 2.1. Demineralization

*Fucus vesiculosus* residues reached stable pH levels of 6 after the water leaching step, a value corresponding to the fresh biomass. This is said to be caused by the macroalgal cell wall polysaccharides that contain acidic functional groups [20]. Four washing steps were necessary to reach pH 7, corresponding to 50 g water/g initial macroalgae. Five washing steps brought the pH level up to 6.9, compared to the initial pH of 5.6 when leaching with HTL water. Citric acid leaching led to the lowest initial pH of 3.8. Eight washing steps only raised the pH to 6.1, corresponding to 100 g water/g initial macroalgae, despite the fact that a diluted acid solution was used.

Ash content was reduced as a result of the dry biomass grinding and subsequent water leaching. A final ash content of 13.71 wt.% was obtained when measured after first reaching pH 7 (i.e., after washing step 4), corresponding to an overall ash reduction of 38.85%. However, the final measurements show that further ash reduction is very limited during neutralization, defined here as 1.14% per four $H_2O$ washes throughout the entire experimental range.

HTL water led to less effective demineralization compared to water leaching. Here, the final ash contents amounted to 16.17 and 14.89 wt.% prior and post neutralization, respectively. The agent's capacity is believed to be limited due to the relatively high amount of inorganics (2.51 wt.%) in the liquid itself. The used catalyst, potassium carbonate, is believed to constitute the majority of the ash and it is hypothesized to add onto the amounts of potassium salts abundant in the seaweed biomass. Neutralization did not offer much in terms of further reductions. The slight reduction amounted to 1.28 wt.%, while an analogous decrease of 1.9 wt.% occurred in the case of water leaching. All in all, diluted HTL water leaching offered inferior ash reduction performance, exhibiting a demineralization potential of 33.59%. The raw data acquired for and used throughout this study are available in the Supplementary Materials.

In the studied fresh seaweed, leaching using a dilute citric acid solution resulted in the highest ash removal efficacy. Final ash contents of 14.53 and 11.85 wt.% were achieved before and after the eight washing steps, respectively. This amounted to ash reductions of 35.19% and 47.15%. The final result is caused by the combined effects of acid and water leaching, as evident from the data. The results of this part of the study are listed in Table 1 and visualized in Figure 1.

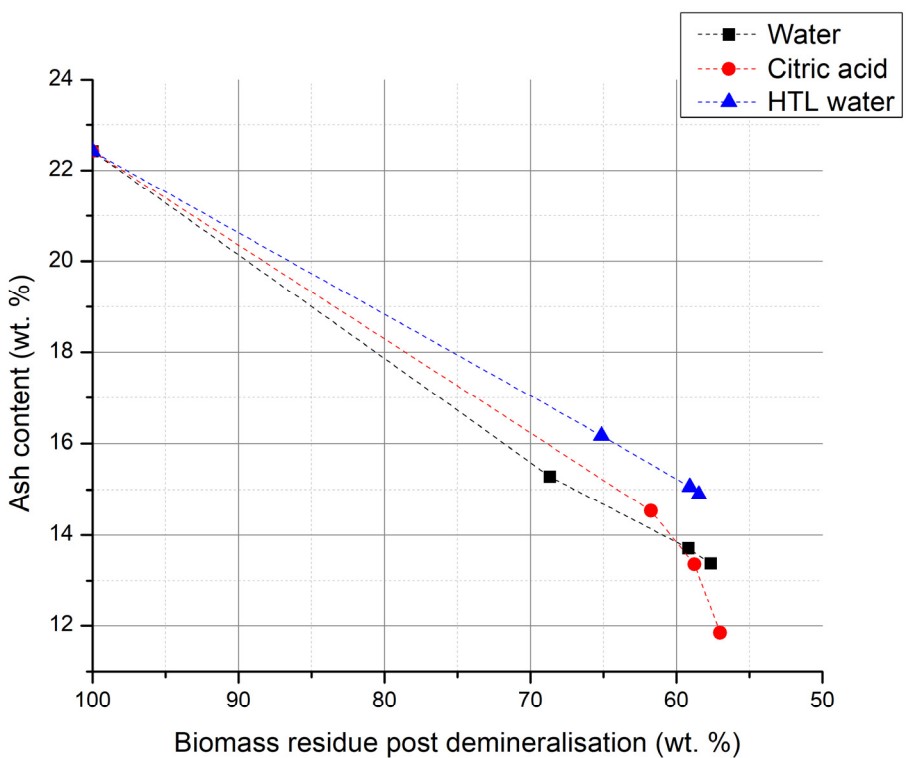

**Figure 1.** Correlation between macroalgae residue and ash content post demineralization.

**Table 1.** Demineralization results, highlighting the effects different leaching agents and post-treatment via water washing have on the pH, final ash content and higher heating value of macroalgal biomass.

| Leaching | No. of $H_2O$ Washes | pH | Ash [wt.%] | Residue HHV [MJ/kg] |
|---|---|---|---|---|
| | 1 | 6.4 | 15.28 ± 0.20 | 15.18 ± 0.08 |
| Water | 4 | 7 | 13.71 ± 0.33 | 15.65 ± 0.05 |
| | 8 | 7 | 13.38 ± 0.36 | 15.46 ± 0.04 |
| | 1 | 5.6 | 16.17 ± 0.91 | 15.73 ± 0.03 |
| HTL water | 4 | 6.7 | 15.06 ± 0.96 | 15.82 ± 0.01 |
| | 5 | 6.9 | 14.89 ± 0.59 | 15.80 ± 0.01 |
| | 1 | 4 | 14.53 ± 0.42 | 16.16 ± 0.02 |
| Citric acid | 4 | 5.3 | 13.36 ± 1.36 | 16.16 ± 0.07 |
| | 8 | 6.1 | 11.85 ± 0.37 | 16.03 ± 0.05 |

Higher heating values (HHVs) of the resultant biomass residues were measured to extend the comparison of the three investigated methods. From an energetic standpoint, higher quality feedstocks were achieved via the use of all three leaching agents. Here, too, the claim that citric acid treatment performs best is valid since the highest HHV recorded was 16.16 MJ/kg compared to the initial 14.95 MJ/kg of the dried *F. vesiculosus* seaweeds.

The final metric here was to measure the amount of solid residues generated post leaching. Differences across the three were observed, with 61.75 wt.%, 65.15 wt.% and 68.66 wt.% of residues generated when leaching with citric acid, HTL water and water, respectively. Thus, the determined differences in ash removal efficacies are confirmed. It is noteworthy that post-treatment water washing equalized the amount of residues by wash number 4 from all three sets of experiments, further suggesting that neutralization and

subsequent washing do not depend on the leaching agent used as far as residue generation is concerned

### 2.2. Value-Added Product Extraction

Despite the high extent of alginate and fucoidan extractions, high quantities of post-extraction residues were produced, amounting to 41.88 and 68.17 wt.% of the initial biomass, respectively.

The residues were exposed to elemental analysis and the effect alginate and fucoidan extractions have on heteroatomic constituents, specifically nitrogen and sulfur, was studied. Fucoidan extraction led to lower amounts of elemental nitrogen, i.e., 1.84 wt.% compared to the initial 3.14 wt.%, but alginate extraction did not affect this parameter of the biomass. Meanwhile, fucoidan extraction resulted in a significantly lower fraction of elemental sulfur. Here, the final content amounted to 0.37 wt.% compared to the initial 1.12 wt.% in the dried seaweeds.

The residues were also studied in terms of how ash content changes after exposure to the two extraction processes. While no significant changes in the final ash content were observed after fucoidan extraction, inorganics were concentrated by the alginate extraction procedure. To confirm, fucoidan and alginate extractives were also ashed. Low amounts of dissolved inorganics were determined in both extracts. Thus, both extraction methods were confirmed to selectively dissolve organics, removing virtually no inorganics. However, the produced ash samples did differ in color (fucoidan extraction residues were light, alginate extraction residues were dark), indicating that of the limited amounts of inorganics removed, the processes do target different compounds. Additionally, as seen in Figure 2, the different residues generated from leaching and extraction did not result in significant differences upon thermal decomposition; divergences are apparent only in the latter stages, when the residue is composed of inorganics to a great extent.

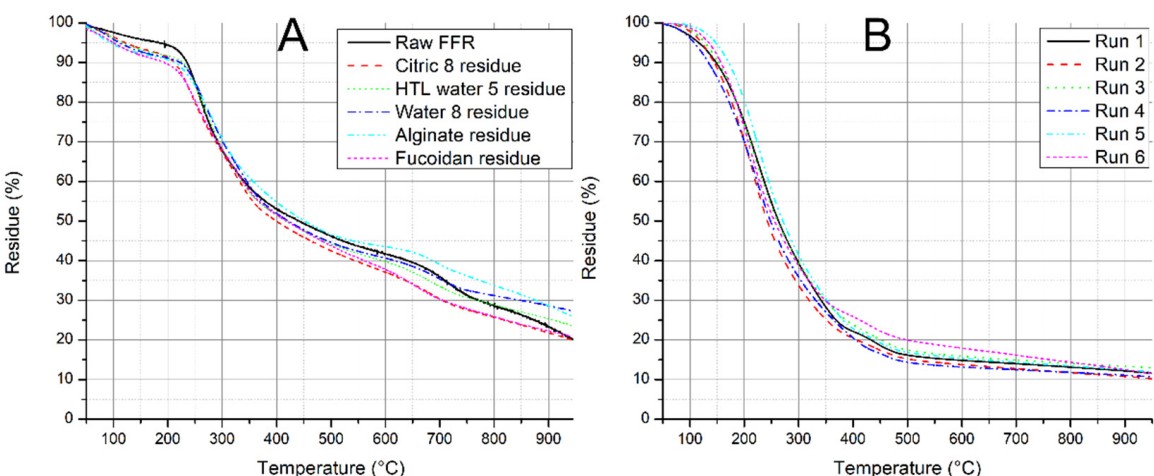

**Figure 2.** TGA curves representing raw biomass, leaching residues, and extraction residues (**A**), and the six produced biocrudes (**B**).

### 2.3. Hydrothermal Liquefaction

#### 2.3.1. HTL Yields

The recorded biocrude yields were in the range of 15.23 to 28.21 wt.% on dry, ash-free (DAF) basis. The results, shown in Table 2, highlight that run 6 (short HTL) and run 5 (HTL of neutralized citric acid leaching residues) resulted in the lowest yields. Run 4 (baseline HTL) produced a slightly higher yield of 19.36 wt.%. Finally, as seen by the results of the three remaining biomass treatment runs resulted in improved biocrude yields. Most notably, run 4 (HTL of alginate extraction residue) experiments yielded the highest quantity of products, amounting to 28.21 wt.%.

**Table 2.** Summary of HTL biocrude yields and key quality parameters, including H/C and O/C ratios and higher heating values and energy recovery rates. DAF basis was used when calculating the biocrude yields.

|  | Biocrude Yield [wt.%] | H/C | O/C | HHV [MJ/kg] | ER [%] |
|---|---|---|---|---|---|
| Run 1 | 19.36 ± 4.48 | 1.38 | 0.35 | 26.28 | 30.95 |
| Run 2 | 21.59 ± 3.73 | 1.22 | 0.10 | 35.35 | 46.93 |
| Run 3 | 26.56 ± 5.34 | 1.29 | 0.13 | 34.46 | 52.83 |
| Run 4 | 28.21 ± 3.44 | 1.36 | 0.13 | 34.50 | 45.78 |
| Run 5 | 17.26 ± 1.1 | 1.55 | 0.08 | 38.05 | 38.39 |
| Run 6 | 15.23 ± 3.82 | 1.54 | 0.05 | 39.16 | 35.29 |

HTL by-product yields were also quantified in terms of mass yields. Runs 2, 3, and 5 generated the most solids. Run 3, representing the thermochemical conversion of fucoidan extraction, yielded the most solids (0.35 g), whereas run 4 (alginate residues) generated the least (0.18 g). Such a significant difference must be taken into consideration, especially when preparing for continuous operation. In terms of water solubles (WS), runs 2 and 3 yielded the least with 0.15 g and 0.09 g, respectively. Here, the overall average amount of produced WS across all experiments was 0.2 g. Run 1 yielded the most: 0.28 g of WS. Except for run 2, the yields of gaseous by-products were comparable across the experimental range. The conversion of citric acid leaching residues generated the most gases: 0.38 g, compared to the average of 0.35 g. Figure 3 shows the yields of all four products, scaled with respect to each other, and represents the proportional yields at lab scale batch processing. As a general tendency, the data suggest that the slightly worse-performing runs generate higher amounts WS and gas, on average.

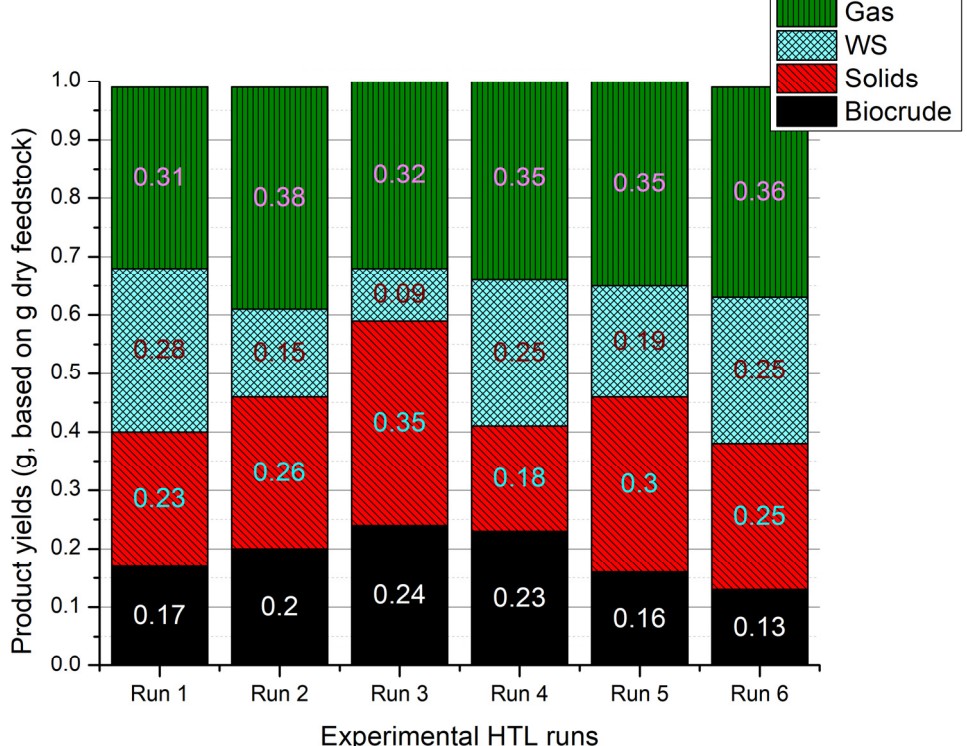

**Figure 3.** Overview of HTL product yields, including biocrude, solids, water solubles and gases, indiscriminately scaled to a closed mass balance.

The biocrudes were also evaluated in terms of their HHVs and the associated calculated energy recovery rates for each set of experimental runs. The produced biocrudes varied greatly in terms of their HHVs, ranging from as low as 26.28 MJ/kg in run 1, up to 38.05 and 39.16 MJ/kg as estimated for run 5 and 6 biocrudes, respectively. The higher-end values are similar to fossil crudes. As a proxy for the feasibility of converting the different biomass streams via HTL, energy recovery levels in the main fuel product were calculated. As per Table 2, the highest ERs were estimated for runs 2, 3, and 4, with the maximum estimated for fucoidan residue HTL being 52.83%. Conversely, run 1 (HTL of untreated seaweeds) performed the worst, reaching just 30.95% in ER. Finally, the energy contents between 35–38% of the initial feedstocks were estimated for the remaining runs 5 and 6.

### 2.3.2. Biocrude Quality

Biocrude quality determination of the different conversion runs is no less important than estimating biocrude yield parameters. Elemental H/C and O/C ratios were the first two indicators. While the goal is to have a biofuel precursor with maximal hydrogen and minimal oxygen amounts, the highest H/Cs of ~1.54 with low O/C ratios between 0.05 and 0.08 were determined in the biocrudes of runs 5 and 6. A post-treatment upgrading step including extensive deoxygenation would still be necessary to approach fossil analogues of high enough quality for commercial refining and blending. Adequate quality parameters were recorded in run 2, 3, and 4 biocrudes. Since only baseline run 1 biocrude was a product significantly more contaminated with oxygen, all pre-treatments/conditions, including demineralization, value-added product extraction, and even short retention, can yield superior biocrudes. Table 3 highlights the key elemental constituents in a succinct manner. No significant differences were observed upon proximate analysis of the different products: volatile matter and fixed carbon averaged at 83.32 ± 2.3 and 16.68 ± 2.3 wt.%, respectively. Here, a high fraction of volatiles is an important parameter indicating the potential suitability for use as a fuel precursor for downstream processing into lighter hydrocarbons such as diesel, jet fuel, and gasoline. TGA analysis revealed concerningly high amounts of inorganic residues, averaging at 11.46 ± 0.7%, as shown in Figure 2, further emphasizing the need for biocrude post-treatment.

**Table 3.** Overview of quantified HTL biocrude sample constituents: carbon, hydrogen, nitrogen, sulfur, and oxygen.

| | C [wt.%] | H [wt.%] | N [wt.%] | S [wt.%] | O [wt.%] |
|---|---|---|---|---|---|
| Run 1 | 60.04 | 6.90 | 3.07 | 1.47 | 28.20 |
| Run 2 | 77.72 | 7.91 | 2.74 | 0.53 | 10.78 |
| Run 3 | 75.28 | 8.07 | 2.96 | 0.49 | 18.88 |
| Run 4 | 74.40 | 8.41 | 3.01 | 0.51 | 13.37 |
| Run 5 | 77.85 | 10.03 | 3.16 | n.m. | 8.65 |
| Run 6 | 79.60 | 10.20 | 4.32 | n.m. | 5.57 |

n.m.—not measured; O calculated by difference, assuming 0.3 wt.% ash.

Differences in biocrude yields and quality between runs 2 and 5 and 1 and 6 can only be done by taking a closer look at the resultant biocrudes; only then is it possible to see whether extensive neutralization or a shorter retention time could hold any advantages. As per Figure 3 and Table 2, biocrude yields of neutralized residue and short retention HTL runs were poor. Only run 5 performed slightly better out of the four. No definitive tendency can be observed from by-product distribution. Out of the two, neutralized residues yielded more gas compared to the leached biomass. Run 1 biocrude had a very high amount of oxygen, as evident in Table 3. Comparing these two, shorter retention seems preferable due to a significantly lower O/C ratio and a slight increase in H/C. When discussing the neutralization extent, however, of a similar O/C ratio, the HTL of neutralized residues

yielded a biocrude with a H/C ratio more than 20% higher than that of non-water washed acid leaching residues.

### 2.3.3. Solids

Organic and inorganic fractions of the generated solid residues were determined and are shown in Figure 4. Averaging at 70.87 wt.%, the organic fractions did not vary significantly across the experimental range. The only exceptions were run 3 (fucoidan residues) solids, of which only 52.54 wt.% were organic in nature. As seen in Figure 3, the amount of gaseous by-product generated during run 3 was average and the yield of WS at the lower end of the spectrum. More inorganics must have been carried through in the solid phase by-product as such a high fraction of the solids was generated. The apparent concentration of organics is, therefore, apparently low. While the ash content of the biomass and HTL solids was measured directly, the missing amount is assigned to the WS fraction, where the corresponding percentage is derived on the basis of the total amount of produced WS. Despite the potential to recycle the water phase, as evidenced in HTL of lignocellulosic biomass, the present study suggests this might not be feasible in HTL of macroalgae nor for demineralization purposes [21]. Upon ICP analysis, the concentration levels of all measured levels except for potassium and sodium were higher in run 1 solids (Figure 5) than those measured in the dried feedstock.

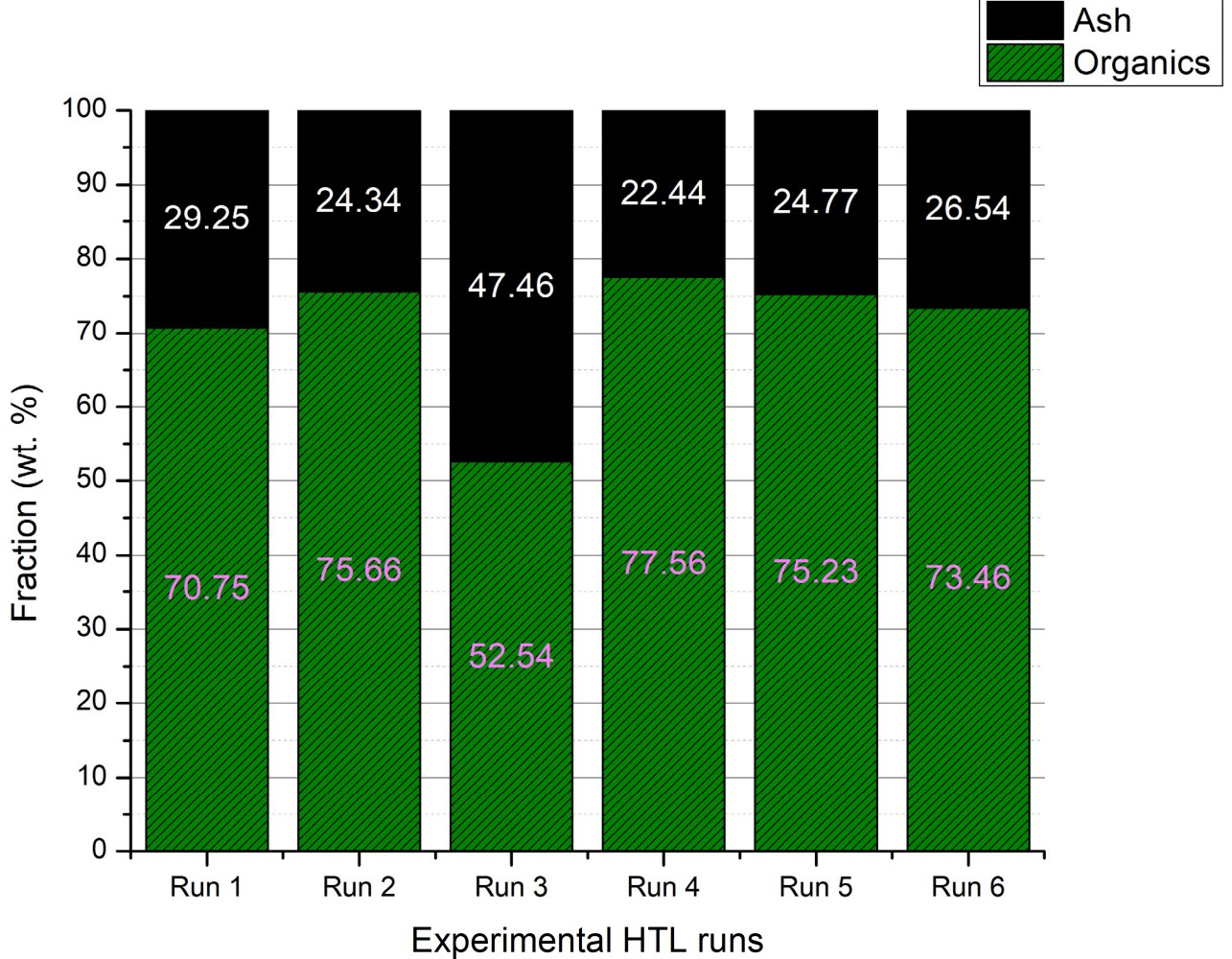

**Figure 4.** Variation in organic and inorganic constituents of the six produced HTL solids.

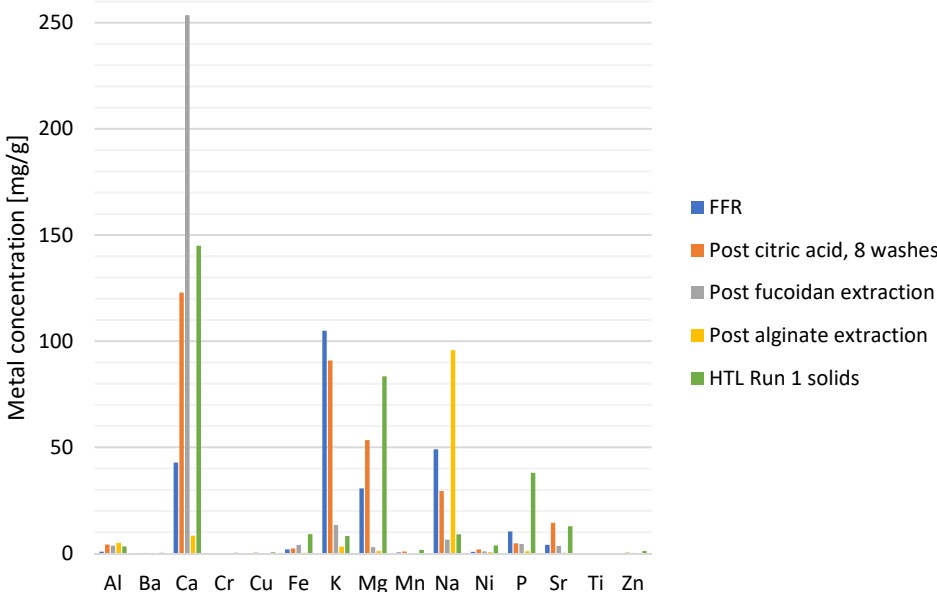

**Figure 5.** Comparative metal concentrations in raw macroalgae, citric acid leaching, fucoidan, and alginate extraction residues, and the solids produced in HTL run 1, as determined by ICP ash analysis.

### 2.3.4. Gases

Similar compositions of constituent gases were determined across the gaseous by-products of the six sets of experimental HTL runs (Figure 6). Runs 3 and 4 differentiated from the other samples. While the concentration of CO was significantly higher in run 3 samples, more $H_2$ was detected in the by-product of run 4. In general, the composition of all product gases was heavily dominated by $CO_2$, with 84.61 and 91.61 vol.% as the minimal and maximal values, respectively. Typically, only trace amounts of CO, $H_2$, and $CH_4$ were detected, confirming that decarboxylation and decarbonylation reactions are behind the removal of oxygen.

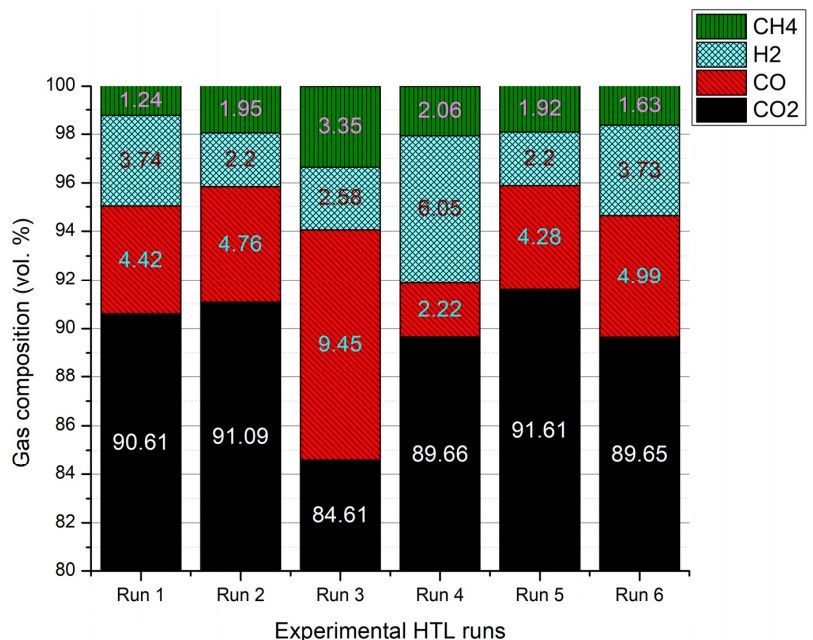

**Figure 6.** Comparative overview of the compositional profiles of the HTL by-product gases across the experimental range.

## 3. Discussion

### 3.1. Demineralization

Should the effect of biomass residue pH be strong and neutrality an important prerequisite for effective HTL of seaweed feedstocks, all demineralization runs would require post-processing neutralization, regardless of the leaching agent that was originally employed. Since it was possible to raise the pH of the HTL water leached residues via water washing, a further synergistic benefit could be acquired by substituting intensive acid pretreatment, given that the leaching agent could be shown to be an effective leaching agent. Here, HTL water usage as a demineralization agent and the subsequent necessity to neutralize the residues deserve further, dedicated studies. There is a potential need to include the use of an external neutralizing agent to render the biomass exposed to citric acid leaching neutral. Alternatively, it is apparent that an alarmingly high water consumption would have to be dealt with downstream. The addition of an alkali catalyst could be synergistically beneficial if both extensive ash removal potential and effective HTL conversion can be shown. However, costs associated with high water consumption and post-treatment are not exclusive to wet acid leaching—each washing step comes at a cost of lost organic matter. This should be taken into account as the neutralization procedure via washing would exacerbate such losses. Reactive solvent citric acid recovery could be a potential way to significantly diminish water demand [22]. While here water washing was studied as a widely available and low-tech method, careful investigations have to be carried out with any potential biomass residue to adequately weigh the pros and cons associated with the presented methods and their extent.

The present study recorded lower HHVs in post-neutralization residues, suggesting that extensive water washing might not be an advantageous method. In fact, the full extent of demineralization is reached before neutralization occurs; thus, any further washes may remove organics more selectively instead. This is confirmed by the elemental analysis data. After citric acid leaching, despite a slightly lower amount of inorganics by 2.68 wt.%, elemental carbon and hydrogen increased only by 1.69% and 0.19%, respectively, when residues after one and eight $H_2O$ washes were compared. This confirms that organics are lost in subsequent washing of macroalgal biomass residues. Finally, the differences in nitrogen were negligible (below 0.1 wt.%), whereas, given the shortage of relevant data, no conclusions can be made about the fate of sulfurous compounds.

It is apparent from the data presented in Table 1 that all three leaching agents lead to higher quality energy feedstocks. Since post-neutralization resulted in decreases in HHVs, it is suggested that extensive water washing might have significant drawbacks. In fact, the highest extent of leaching is seen to take place during the first washing steps, as shown by the average biomass mass losses of 34.82% and 40.98% recorded after the first and fourth wash, respectively. Figure 1 depicts the change in ash content versus biomass residue. As an example, here, a decrease in ash content by 31.85% (i.e., a 7.14% mass loss of the biomass) results in a total biomass mass loss of 31.34%. This indicates that the tested methods are several times more effective at removing organics. Costs associated with feedstock are typically the limiting barrier for biofuel production via HTL and this is no exception for macroalgae [23]. Whether the advantages of demineralization would outweigh the inherent additional costs of processing and leached organics valorization is yet to be shown quantitatively.

Highly selective demineralization of biomass high in inorganics is a topic of high interest in the scientific community. More aggressive acidic de-ashing, such as the use of nitric acid, was shown to be highly effective but still lacked in selectivity [24]. Meanwhile, novel methods, such as treatment with pulsed electric fields, have already been used to reach high demineralization efficiency at a relatively low loss of organics [25]. More importantly, however, it was recently shown that batch aqueous demineralization overestimates water-soluble inorganics and loss of organic matter, indicating the urgent need for large scale semi- or fully continuous pretreatment technique research on the biomass of interest [26]. This seems to be true also for biocrude ash content determination,

where even low ash (1–2% db) feedstocks lead to 4–5% ash in continuous supercritical HTL biocrudes [27]. This highlights the possibility that achieving high demineralization extent would not necessarily prevent the need to purify the produced biocrude downstream, as the inorganic homogeneous catalysts commonly used in HTL (e.g., NaOH, $K_2CO_3$) partially migrate to the product stream. It is the authors view that research is needed in continuous demineralization of biomass high in inorganics and subsequent HTL to further investigate whether it is more economical to develop efficient demineralization treatments or HTL systems capable of processing high ash feedstocks instead.

### 3.2. Value-Added Product Extraction

The high amounts of residues could be explained by the fact that the *F. vesiculosus* used in the present study had been harvested in late winter, suggesting that the plants had consumed their energy stocks, giving rise to the high amount of inorganics present in the feedstock. Further differences have been recorded when comparing winter-harvested *F. vesiculosus* and autumn-collected *S. latissima*, where the latter contained more than twice more alginate [28,29]. Obviously, genera-specific differences in plant structures could also add to such differences.

The apparent variance in residue coloration served as a solid basis for further ash analysis to identify how this visual difference correlates to compositional differences. Calcium and sodium compounds made up the majority of the total inorganics in fucoidan and alginate extraction residues, respectively. Since $CaCl_2$ and $Na_2CO_3$ cannot completely be removed in a single post-extraction washing step, significantly increased concentrations of these elements are observed upon ICP analysis. The lightness of the fucoidan residues is a result of the high concentrations of calcium in the biomass. The inorganic constituents of the dried *F. vesiculosus* itself are dominated by alkali and alkali earth metals. Specifically, potassium, sodium, calcium, and magnesium are abundant in the highest amounts, while potassium, magnesium, and manganese are shown to be extracted effectively throughout both of the procedures. However, heavier metals, such as aluminum, copper, nickel, and zinc, remain at nearly unchanged concentrations. Particular to alginate extraction is the enhanced migration of iron, strontium, and phosphorous. This phenomenon is explained by the use of $H_2SO_4$—this strong acid is known to be a more aggressive demineralization agent [30]. When compared to the initial macroalgae, the HHVs of both extraction residues exhibited improved HHV values.

### 3.3. Hydrothermal Liquefaction

As reported previously, batch macroalgal biocrude can contain a significant quantity of inorganics, rendering further purification necessary [31]. Several techniques, including filtration, electrocoalescence, washing, and, more recently, washing with carbonated water, utilized in conventional refineries could be employed for this purpose [32,33]. The concentrations of alkali and alkaline earth metals were expected to change due to the high solubility of potassium and sodium salts in water—significant amounts were removed during the product separation procedure, as observed previously in the literature [34]. ICP analysis of the inorganics present in the produced by-product solids indicated that reactor degradation is an area of concern and further studies are necessary to determine specific degradation rates. This is based on the increasing or appearing concentration levels of stainless steel-derived metals, such as chromium, iron, manganese, nickel, and titanium.

Yet another important methodological detail worth noting is that an additional water washing step was included after product filtration to check for significant amounts of water solubles that had precipitated upon rinsing the reactants with acetone. Obviously, using a solvent to empty the reactors and separate the products will impact the results to some extent and the obtained products may not be representative of larger-scale operations. This is made especially clear as gravimetrical separation of biocrude is commonly employed at continuous pilot-scale HTL facilities [21]. Largely varied mass losses between 7.96 to 54.15% were recorded upon the additional water washing step in run 4 and 3 solids, respectively.

The recalcitrance of alginate extraction residue derived HTL solids is suggested to be brought on, once again, by the use of sulfuric acid that had made the residues more stable hydrolytically via structural destruction and effective demineralization. In a previous study, crystalline macroalgae structures were shown to be broken down by dilute sulfuric acid in *L. digitata* [31]. This analysis revealed no effect on the amount of acetone WS precipitates by extensive water neutralization or shorter HTL retention time as mass losses of 37 and 22 wt.% were recorded in solids of runs 1 and 6, and 2 and 5, respectively. Nonetheless, given the large variation and possible high misrepresentation of the yields, extra washing makes sense to show quantitively certain data, and reduce the risk of them not being either acetone- and water-insoluble, or representative of large HTL facilities filtered in-line.

## 4. Materials and Methods

Fresh samples of brown macroalgae *Fucus vesiculosus* (photograph in Figure 7), growing north of the Danish mainland, were acquired for the experimental part of the study. Specifically, these algae were chosen due to their wide distribution in the Baltic Sea. In some western areas, *F. vesiculosus* are the only large, canopy-forming brown macroalgae. They grow along rocky coasts, at low depths [35].

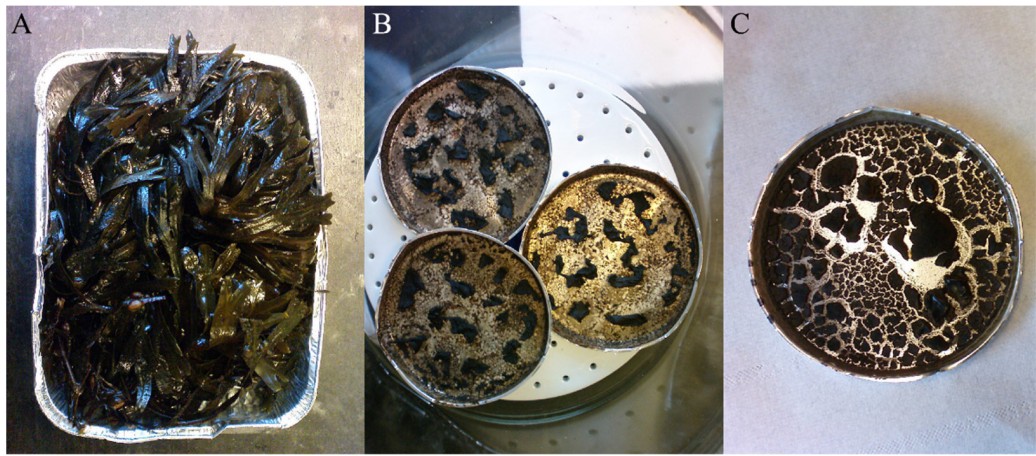

**Figure 7.** Physical appearance of the samples: fresh biomass (**A**), dried alginate extraction residues (**B**), and dried fucoidan extraction residues (**C**).

Thermogravimetric analysis (TGA) was performed in an inert atmosphere (purged with nitrogen) using a PerkinElmer STA6000 TG/DSC analyzer. Samples of 4–7 mg were heated to 950 °C at a temperature ramp rate of 10 K/min. The nitrogen flow rate was set to 20 mL/min throughout the entire procedure. CHNS analysis was carried out on a Vario Macro Cube simultaneous CHNS analyzer from Elementar. Here, samples of 70–80 mg were analyzed in triplicates. An in-house moisture analysis (KERN MLS) was used to determine the water content in the fresh biomass. Higher heating values (HHVs) of the dried and milled macroalgae samples were measured in triplicates using an IKA C2000 basic bomb calorimeter. Finally, ash content in the biomass samples was determined as the constant mass solid residue post-dry oxidation at 575 ± 25 °C. All compositional information is provided in Table 4.

**Table 4.** Compositional analysis of freshly harvested winter *F. vesiculosus* macroalgae.

| | |
|---|---|
| Physical Properties (as Received): | |
| Water content [%] | 77.42 ± 0.5 |
| Proximate Analysis (Dry Basis): | |
| Volatile matter [%] | 54.16 |
| Fixed carbon [%] | 20.23 |
| Ash [%] | 22.42 |
| Higher heating value [MJ/kg] | 14.95 ± 0.01 |
| Elemental Analysis (Dry, Ash-Free Basis): | |
| Carbon [%] | 36.90 |
| Hydrogen [%] | 6.06 |
| Nitrogen [%] | 3.14 |
| Sulfur [%] | 1.12 |
| Oxygen [%] [a] | 30.36 |

[a]—calculated by difference.

### 4.1. Demineralization

Initial screening tests were carried out on *Laminaria digitata* brown macroalgae as a part of a previous study [30]. The combination of significant ash removal and relatively water-lean neutralization procedure led to evaluating dilute citric acid treatment as optimal.

Demineralization with distilled water was also carried out to establish baseline results. Finally, to investigate an alternative means to utilize one of the by-product streams of continuous HTL, the aqueous phase by-product was used as the third leaching agent. Its acidic nature gives merit to investigate the de-ashing potential, and thus, valorize the otherwise challenging waste stream. The raw aqueous product was a sample previously collected at the local semi-continuous HTL plant and represents a real-world synergistic opportunity. The sample was slightly acidic with a pH level below 5.5 [27].

The raw macroalgae were pre-rinsed with cold water to remove any unbound inorganics as the first step. After the initial rinsing, the biomass was oven-dried and milled (FOSS CyclotecTM 1093, particle size: ≤200 μm). In the case of citric acid leaching, the now dry and powdered macroalgae were mixed with a 1 wt.% citric acid solution (12.5 g solution/g macroalgae). The leaching process took place overnight (18 h of continuous stirring at 1000 rpm at room temperature). After leaching, the mixtures were centrifuged (SIGMA 6–16S centrifuge, for 5 min at 4000 rpm) to remove the leachate. Then, the neutralization/rinsing procedure took place. Neutralization is a part of the study to process a non-acidic feedstock. This was done because alkaline processing media were found to suppress char formation from carbohydrates during HTL [21]. Distilled water was added to the residues (12.5 g water/g initial macroalgae) and the mixture was stirred manually. Subsequent centrifugation was utilized for separation. Varying amounts of coupled rinsing-separating steps were enforced to set up for analysis of HTL of post-demineralization macroalgae. The focus here was to determine whether a great neutralization extent is truly necessary for efficient HTL of acid leached macroalgae. The experimental design included drying (at least for 18 h at 105 °C) the residues after one, four, and eight rinsing repetitions (i.e., simulated water consumption ranging from 12.5 to 100 g/g of dry initial macroalgae). Ultimately, we aimed to test the need for water-intensive post-treatment. The pH levels were measured initially, after the leaching period and after each rinsing step (WTW pH 3210 m, accuracy of ±0.2 pH points). All results are reported as average values of triplicate experiments/measurements, unless stated otherwise.

Just 50 mL of HTL aqueous phase was available for the needs of this study. In order to accommodate the required leaching medium, the available 50 mL were diluted with distilled water to reach a total volume of 300 mL. This being said, it is worthwhile to note

that the pH of the solution did not change significantly, stabilizing at pH 5.6 prior to mixing. The same acid solution-to-biomass ratio of 12.5 and leaching conditions were kept.

### 4.2. Value-Added Product Extraction

The experimental flow of the performed alginate removal procedures was adapted from [36]. Three samples (sample size: 5 g) of the winter harvest *F. vesiculosus* were processed. Firstly, the rinsed macroalgae were dried and milled. Then, the powder was mixed with a 0.5 M $H_2SO_4$ solution (13.58 g solution/g algae) and stored overnight (minimum 21 h) in a dark cabinet. Then, the mixture was centrifuged (5 min at 4000 rpm) and the liquid solution was removed. An intermediary washing step (13.58 g $H_2O$/g initial algae) with subsequent centrifugation (4000 rpm, 5 min) was performed to remove any residual acid. A 4% $Na_2CO_3$ solution (19.95 g $Na_2CO_3$ solution/g initial algae) was added to the residues. The mixture was stirred magnetically (800 rpm) for 2 h. After soaking, the mixture was once again centrifuged to separate the solubles. A washing step (19.95 g water/g initial algae—mix, centrifuge, drain) took place next. All of the above process steps were carried out at room temperature. Finally, the residues were carefully removed from the centrifuge bottles and placed in an induction oven to dry for at least 18 h at 105 °C.

The employed simulative fucoidan extraction procedure was adapted from [15]. Three samples (sample size: 5 g) of winter harvest *F. vesiculosus* were used. The water-rinsed macroalgae were processed mechanically via drying and milling. Subsequently, fucoidan was extracted in a $CaCl_2$ solution. The extraction was finished throughout two steps: samples were exposed two times to 20 min-long magnetic stirring (800 rpm) sessions in 1 wt.% $CaCl_2$ solutions (16.67 g solution/g algae). After each stirring, the mixtures were centrifuged at 4000 rpm for 5 min and the separated liquid was removed. A similar procedure followed the two extraction-separation steps: the solid fucoidan extraction residues were mixed with water (16.67 g water/g initial algae) and centrifuged once more in order to remove any remaining calcium chloride. All steps were carried out at room temperature. Finally, the residues were oven-dried, cooled in a desiccator, weighed, and stored in airtight containers until further processing.

### 4.3. Hydrothermal Liquefaction

Six separate HTL runs were carried out throughout this study. The experiment list can be seen in Table 5. The main focus of the overall procedure was set on HTL of treated macroalgae, namely de-ashed, post fucoidan extraction and post alginate extraction. Additionally, the effects of post de-ashing neutralization and a shorter retention time were investigated.

**Table 5.** HTL experimental overview.

| Reference | Pre-Treatment | HTL Conditions | Hypothesis/Argument |
|---|---|---|---|
| Run 1 | - | | Baseline |
| Run 2 | De-ashing | | Demineralization improves yield |
| Run 3 | Fucoidan extraction | Normal | Effective HTL with fucoidan extraction residues is possible |
| Run 4 | Alginate extraction | | Effective HTL with alginate extraction residues is possible |
| Run 5 | De-ashing and neutralization | | Post de-ashing neutralization is not necessary |
| Run 6 | - | Short | High-quality biocrude can be produced at a shorter reaction time |

All experiments were carried out in stainless steel (grade 316) 12 mL microreactors. Feedstock dry mass loadings of 20% were used, and all reactions were carried out at 400 °C (±5 °C). Upon feedstock slurry preparation, the specific macroalgae powder was

combined with distilled water to form the predefined mixture. A total of 5 g (±0.1 g) of the slurry was then loaded into the reactors. Nitrogen gas was used to simultaneously leak test (80 bar) and purge the reactors to evacuate atmospheric oxygen. Hereafter, two reactors and thermocouples were mechanically coupled to an agitator, providing mechanical mixing of the reagents inside the reactors while being processed. The two reactors were then submerged into a preheated, fluidized sand bath and held, normally, for 15 min of retention time and 10 min in the short HTL run. The retention time was defined as the time that passes between the moment when the reactors have reached the pre-set temperature of 400 °C (±5 °C) and the instance of manually submerging the reactors into the cool (~20 °C) water bath. After quenching in water for a minimum of half an hour, the separation procedure begins.

The first step of product separation was the weighing of the gaseous products, gas sampling, and venting the remaining gases via top-mounted valves. The remaining products consisted of solid residues, biocrude, and an aqueous phase. The reactors were washed with acetone to remove all biocrude traces from the reactor. The liquid phase was then separated from the char via vacuum-assisted mechanical filtration (VWR, particle retention: 5–13 μm). The solids present on the filter were then dried overnight at 105 °C and re-filtered with 250 mL of distilled water. The remaining solid residues were dried once again, weighed, and defined as water and acetone insoluble solids. Finally, the produced solids were ashed. This was done to determine how much inorganics are present in the by-product. Acetone was then evaporated from the homogeneous liquid fraction and the biocrude fraction was manually extracted after centrifuge-aided phase separation. The higher density extracts were defined as biocrude, whereas the aqueous by-product was collected, dried, weighed, and denoted as water solubles (WS). Post reaction gases were weighed, adjusted for initial nitrogen addition, and analyzed via GC analysis. The aqueous products were weighed prior and after to show the extent of experimental error due to water losses during acetone evaporation. The produced biocrudes were weighed, their proximate analyses were done via TGA, and their water contents were measured via Karl Fischer titration.

### 4.4. Calculation Methods

This section describes all calculative methods that were used for determining both product/by-product yields and quality parameters, such as biocrude higher heating value (HHV), hydrogen-to-carbon ratio (H/C), and oxygen-to-carbon ratio (O/C). Both biocrude and gas yields were calculated on a dry and ash-free (DAF) feedstock basis identically as shown in Equation (1). Similarly, the yields of solids were calculated on a dry basis.

$$Yield_{biocrude} = \frac{Mass\ of\ biocrude}{Mass\ of\ dry,\ ash\ free\ feedstock} \cdot 100\% \tag{1}$$

Finally, the yields of WS were determined on a dry feedstock basis, by adding the weighed WS and the amount of solids washed out with water (Equation (2)). This procedure was adopted to better represent the generated amount of WS. Previously utilized methods of presenting the data as process water + WS were shown to be inconsistent (i.e., variations in mass up to 25% among single run triplicate data). Such differences are believed to be caused by the non-automated evaporation step—depending on the duration of this step, more or less process water is lost. However, this does not impair the results of the study as preserving process water was never among the objectives. Furthermore, presenting dry WS data instead is more reliable.

$$Yield_{WS} = \frac{Mass\ of\ WS + mass\ lost\ during\ water\ washing\ of\ solids}{Mass\ of\ dry\ feedstock} \cdot 100\% \tag{2}$$

Due to the inability of measuring HHVs of the produced HTL biocrudes directly (microreactors do not yield sufficient amounts), the study resorted to elemental HHV estimation. In order to represent the biocrude comparably, several HHV estimation formulas

were tested against laboratory measurements or raw macroalgae, demineralization, and value-added product extraction residues. The correlation derived by Friedl et al. (Equation (3)) was shown to give the most accurate results—all tested values were within 5% of the experimental measurements [37]. Meanwhile, the correlation proposed by S. Channiwala and P. Parikh (Equation (4)) was used when estimating the HHVs of the produced biocrudes [38]:

$$HHV = 0.00355 \cdot C^2 - 0.232 \cdot C - 2.230 \cdot H + 0.0512 \cdot C \cdot H + 0.131 \cdot N + 20.6 \qquad (3)$$

$$HHV = 0.3491 \cdot C + 1.1783 \cdot H + 0.1005 \cdot S - 0.1034 \cdot O - 0.0151 \cdot N - 0.0211 \cdot Ash \qquad (4)$$

Hydrogen-to-carbon (H/C) and oxygen-to-carbon (O/C) ratios, on an elemental basis, were calculated for each of the produced biocrudes. Here, analyzed sample masses are taken into account. Such quality parameters allow for a direct comparison with biocrudes produced from other biomass sources, different HTL conditions, and even fossil fuels. In the literature, yet another ratio, the effective hydrogen-to-carbon ratio, is often presented to compensate for any water present in the produced biocrude. Contrary to such an approach, the study included measuring the total water content by Karl Fischer titration and subtracting the results both from biocrude yields and elemental composition.

Additionally, to biocrude yield and quality, energy recovered in the form of produced biocrudes was calculated as well to compare the energetics of each HTL run. The recovered ratio is calculated on dry feedstock basis, using Equation (5):

$$ER = \frac{Mass\ of\ biocrude \cdot estimated\ HHV}{Mass\ of\ dry\ feedstock \cdot measured\ HHV} \cdot 100\% \qquad (5)$$

## 5. Conclusions

All three of the studied demineralization agents led to lower amounts of ash in the macroalgal biomass residues, and correspondingly, superior higher heating values compared to the initial seaweed feedstock. The amounts of generated solid residues leveled out already after the fourth washing step. Citric acid leaching with extensive neutralization was the most effective method for reducing ash, corresponding to a final ash removal efficiency of 47.15%. Ash concentration was observed in biomass residues after alginate extraction. Both alginate and fucoidan extraction residues resulted in improved high heating values.

The recorded hydrothermal liquefaction yields of 26.56 and 28.21 wt.% on a dry ash-free basis were the highest across the experimental range and were obtained when converting fucoidan and alginate extraction residues, respectively. Short retention and neutralized leaching residue experiments exhibited the poorest liquefaction efficiencies. However, these two experiments yielded biocrudes of the highest H/C and lowest O/C ratios, while the baseline dried *F. vesiculosus* product had a higher O/C ratio.

Demineralization led to a slight improvement in biocrude yield. Both fucoidan and alginate extraction residue conversions resulted in relatively high biocrude energy recovery rates of 52.83 and 45.78%. Post macroalgae leaching neutralization is not advised, as it led to significantly poorer biocrude yield and energy recovery rate, and high process water demand. Short retention time hydrothermal liquefaction of *F. vesiculosus* resulted in a low biocrude yield and a high elemental nitrogen content in the product, but promises high quality otherwise, as suggested by advantageous H/C, O/C ratios and a higher heating value of the biocrude.

**Supplementary Materials:** The following are available online at https://www.mdpi.com/article/10.3390/recycling6030045/s1, MS Excel Workbook: Raw data.

**Author Contributions:** Conceptualization, L.J. and T.H.P.; methodology, T.H.P. and L.A.R.; data curation, L.J.; writing—original draft preparation, L.J.; writing—review and editing, T.H.P. and

L.A.R.; visualization, L.J.; supervision, T.H.P. and L.A.R. All authors have read and agreed to the published version of the manuscript.

**Funding:** This research received no external funding.

**Institutional Review Board Statement:** Not applicable.

**Informed Consent Statement:** Not applicable.

**Data Availability Statement:** The data presented in this study are available in the Supplementary Workbook, S1.

**Conflicts of Interest:** The authors declare no conflict of interest.

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
