# Peer review of "Biocrude Production via Non-Catalytic Supercritical Hydrothermal Liquefaction of Fucus vesiculosus Seaweed Processing Residues"

_recycling, doi:10.3390/recycling6030045_

Round 1
Reviewer 1 Report
Journal
Recycling (ISSN 2313-4321)
Manuscript ID: recycling-1225169
Title
Biocrude production via non-catalytic supercritical hydrothermal liquefaction of Fucus vesiculosus seaweed processing residues.
Review
The work presented experimental research done on macroalgae and their utilization to have biocrude oil through the process of hydrothermal liquefaction. It is the opinion of this reviewer that the work done deserves to be published. However, some parts of this manuscript could/should be improved for a more clear explanation of the methods adopted. My comments are only meant to improve the article and no critics are done on the work in general.
This research is very interesting and with few modifications, the paper will be great.
More detailed comments
- Line 12 to line 15. The sentence “Acid leaching using 3 agents…..” is a bit confusing. Better to use short sentences and not a passive form.
- Line 104. Authors could describe more what kind of residue is used from the industrial process. If this info can be disclosed.
- The introduction could introduce the methods used in this research and what is the added value of this particular research in regards to previous research. Maybe the outline of the work could be described by a diagram.
- HTL could be more suitable than SCWG in regards to safety issues, however could also present drawbacks. This could be explained more into the introduction section, this is because one question could be: why HTL and not SCWG? One example of a study of safety issues and SCWG can be found in the paper: Energies 2021, 14(10), 2863; https://doi.org/10.3390/en14102863.
- Results are in section 2, should it be first Materials and Methods and then the results? But maybe the journal requires this format.
- Line 426. From where exactly the samples are taken? Is that possible to have a photo? (not really necessary but it would help).
- The central part that could/should be improved is the description of the pre-treatment before the HTL process. This is because it is particularly interesting. For example, Line 501, the residue refers to which part exactly? The steps are so many that it would be nice to have some diagrams to better explain the steps involved. The word residue is used often, what I mean is to clarify the steps.
- It would be beneficial to have a photo of the equipment used for the HTL process.
- Line 593-595, could be in the discussion.
- The conclusion part is quite short. It would be beneficial to have some consideration in regards to process integration and the feasibility of the process with focus on future prospects.
Reviewer 2 Report
Manuscript Number: recycling-1225169
Title: Biocrude production via non-catalytic supercritical hydrothermal liquefaction of Fucus vesiculosus seaweed processing residues
The work was focused on process the residue by HTL after the extraction of some high-value products from a seaweed. The feedstock was a seaweed (Fucus vesiculosus), from which alginate and fucoidan were extracted, also reduces the remaining ashes in the materials, although in not a great extent. In terms of the biocrude yield, the best result was obtained when citric acid was used and the best ER after fucoidan extraction.
Recommendation:
Accepted with minor corrections.
Comments to the authors.
Major comments
There is no description about the experimental procedure to determine ash content.
Starting from the 2.3 Section, the authors talk regarding runs 1-6 without an explanation about what each test is about, at least a reference to table 5 is need.
Please explain the relation between: a) Abstract -line 16- and conclusions “with citric acid leaching with extensive neutralisation reaching the highest ash removal efficiency of 47%., plus “Final ash contents of 14.53 and 11.85 wt. % were achieved prior and post 8 washing steps…” (section 2.1 , lines 142-143). And b) “The amounts of generated solid residues levelled out already after the fourth washing step (Conclusions: lines 595-596).
In the DTA test you use N2; Is the residue really inorganic? explain
Minor comments
Line 57. what are the properties the authors refer to?
Round 2
Reviewer 1 Report
The authors provided suitable answers to all the questions and comments presented.
Just a few added comments:
- in equations, I suggest using dots instead of "*" and shorter terms or abbreviations. Then I would explain the terms within the text or in a table.
- A photo of the reactors would also be beneficial.